# A flexible and physically transient electrochemical sensor for real-time wireless nitric oxide monitoring

Rongfeng Li[1,7], Hui Qi[2,7], Yuan Ma[3,7], Yuping Deng[1,7], Shengnan Liu[1], Yongsheng Jie[2], Jinzhu Jing[4], Jinlong He[5], Xu Zhang[5], Laura Wheatley[6], Congxi Huang[1], Xing Sheng[3], Milin Zhang[3] & Lan Yin [1✉]

Real-time sensing of nitric oxide (NO) in physiological environments is critically important in monitoring neurotransmission, inflammatory responses, cardiovascular systems, etc. Conventional approaches for NO detection relying on indirect colorimetric measurement or built with rigid and permanent materials cannot provide continuous monitoring and/or require additional surgical retrieval of the implants, which comes with increased risks and hospital cost. Herein, we report a flexible, biologically degradable and wirelessly operated electrochemical sensor for real-time NO detection with a low detection limit (3.97 nmol), a wide sensing range (0.01–100 μM), and desirable anti-interference characteristics. The device successfully captures NO evolution in cultured cells and organs, with results comparable to those obtained from the standard Griess assay. Incorporated with a wireless circuit, the sensor platform achieves continuous sensing of NO levels in living mammals for several days. The work may provide essential diagnostic and therapeutic information for health assessment, treatment optimization and postsurgical monitoring.

[1] School of Materials Science and Engineering, The Key Laboratory of Advanced Materials of Ministry of Education, State Key Laboratory of New Ceramics and Fine Processing, Center for Flexible Electronics Technology, Tsinghua University, Beijing 100084, China. [2] Laboratory of Musculoskeletal Regenerative Medicine, Beijing Institute of Traumatology and Orthopaedics, Beijing 100035, China. [3] Department of Electronic Engineering, Beijing National Research Center for Information Science and Technology and Beijing Innovation Center for Future Chips, Tsinghua University, Beijing 100084, China. [4] Animal Center, Beijing Institute of Traumatology and Orthopaedics, Beijing 100035, China. [5] Tianjin Key Laboratory of Metabolic Diseases, Department of Physiology and Pathophysiology, Tianjin Medical University, Tianjin 300070, China. [6] Trinity College, University of Oxford, Oxford OX1 3BH, UK. [7] These authors contributed equally: Rongfeng Li, Hui Qi, Yuan Ma, Yuping Deng. ✉email: lanyin@tsinghua.edu.cn

Precise and continuous measurements of critical biomarkers in the human body form an important basis for health assessment, pharmaceutical guidance, surgical intervention protocols, and postsurgical monitoring. Specifically, real-time monitoring of nitric oxide (NO) levels in physiological environments plays an essential role in neurotransmission, immune responses, cardiovascular systems, angiogenesis, microcirculation, etc.[1,2]. Abnormal amounts of NO have been reported to be closely associated with inflammation, neurovirulence and cancer progression[3,4]. For example, chondrocytes in osteoarthritis patients are associated with increased inducible NO synthase (iNOS) leading to significant NO generation, which promotes inflammatory responses, chondrocyte apoptosis, and cartilage degradation[5,6]. As one of the leading causes of disability, osteoarthritis is expected to impact at least 130 million individuals globally by 2050. Therefore, probing for NO in the articular cavity could be of great importance for early intervention and treatment optimization of osteoarthritis patients[7]. However, it remains a great challenge to precisely capture the NO concentration in physiological environments due to its short half-life (6–10 s), low concentration (nM–μM), high chemical activity and interference by other chemicals (e.g., glucose, nitrites, and uric acid) in biological systems[8,9]. Several techniques have been proposed to detect NO concentrations, including indirect methods such as Griess assays that measure the concentration of nitrite ion ($NO_2^-$) in solutions and direct methods such as fluorescent probes, electron spin resonance spectroscopy, and chemiluminescence[10–13]. Most of these techniques either suffer from insufficient detection limits or involve complicated sample preparation that impedes real-time measurements of NO in physiological environments[14,15]. By contrast, electrochemical sensors fabricated through a cost-effective process can offer fast and continuous NO detection with high sensitivities and low detection limits[16,17]. However, most conventional electrochemical NO sensors are made of rigid materials and require surgical retrieval if implanted to eliminate unnecessary device loads, which could cause significant irritations and expose patients to infection complications[18,19]. Recently, an emerging class of flexible and physically transient device systems, which has mechanical properties matching those of biological tissues and can be resorbed or physically disappear to benign end products, holds the potential to address the above disadvantages, by reducing potential foreign body and inflammatory responses and eliminating a second surgery for device retraction[20,21]. Remarkable examples, include biodegradable and bioresorbable electronic devices capable of monitoring the pressure and temperature in the brain[22], recording the pressure and strain of tendon healing[23], and spatiotemporally mapping the electrical activity on the cerebral cortex[21]; furthermore, there are bioresorbable therapeutic devices for cardiovascular diseases[24], peripheral nerve regeneration[25], and infection abatement[26].

Although various transient devices have been obtained with desirable performances on a par with the non-transient counterparts, the development of devices with chemical sensing capability in physiological environments is still challenging, as it remains problematic to simultaneously satisfy accurate sensing performance and degradability. Reported strategies to prolong the stability of biodegradable devices include utilizing encapsulation layers and/or electrode materials with slow degradation rates (e.g., molybdenum and highly doped silicon), which however might not guarantee sufficient stable performance or require complicated fabrication processes[27]. In addition, most reported encapsulation methods involve coating of materials on sensing electrodes that might not apply for devices that need direct exposure to chemicals of interest[25,28].

Herein, we demonstrate materials strategies, device architectures and fabrication schemes to achieve a flexible and degradable electrochemical sensor capable of NO detection with a low detection limit (3.92 nM), a wide sensing range (0.01–100 μM), a high temporal resolution (<350 ms), and desirable anti-interference characteristics. Real-time monitoring of NO is demonstrated successfully not only at the cellular and organ levels but also in the joint cavity of a rabbit for a 5-day period with a wireless data transmission system (Fig. 1a). The device is capable of complete physical transience both in vitro and in vivo through potential hydrolysis, disintegration, phagocytosis, and metabolic clearance processes. Biocompatibility assessments show no significant adverse effects or accumulation of foreign materials at implantation sites or in major organs. These results establish important routes toward flexible and biodegradable NO sensing with accurate and stable characteristics in physiological conditions providing essential diagnostic and therapeutic information.

## Results

**Materials synthesis and device fabrication.** A schematic illustration of the transient NO sensor appears in Fig. 1a, and the corresponding fabrication procedure is given in Supplementary Fig. 1. The device consists of a bioresorbable substrate (copolymer of poly(L-lactic acid) and poly(trimethylene carbonate), PLLA–PTMC), ultrathin gold (Au) nanomembrane electrodes, and a biocompatible poly(eugenol) film as the selective membrane (Fig. 1a).

The PLLA–PTMC copolymer substrate is highly flexible and stretchable and is biodegradable through hydrolysis[29,30]. Patterned ultrathin Au nanomembranes (thickness ~32 nm) serve as the working electrode (WE), the counter electrode (CE), and the reference electrode (RE), allowing sensing stability and eventually complete transience through potential disintegration, phagocytosis, and metabolic clearance processes[31]. Biocompatibility and degradation studies of Au nanomaterials are mostly focused on nanoparticles, which have been proposed for various biomedical applications such as chemoradiation, photothermal therapy, drug delivery, etc[32–34]. Despite of some contradictions, many reports suggest that gold nanoparticles are nontoxic with proper size and dosage, and metabolization occurs through kidney, bladder, or hepatobiliary systems[35–37], e.g., no significant side effects have been observed after 24 h with intravenous injection of gold nanoparticles (~100 μg)[38]. Although Au has been considered to be chemically inert, recent study reveals that gold nanoparticles (4–22 nm) are degraded in vitro by cells which are mediated by nicotinamide adenine dinucleotide phosphate oxidase in the lysosome, followed by a recrystallization process, indicating a potential metabolization mechanism for a trace amount of gold[39]. Poly(eugenol) layer (thickness ~16 nm) is incorporated to promote sensing selectivity and specificity toward NO by hydrophobic repulsion, ionic interaction, and molecular exclusion[40,41]. Eugenol, the main chemical component of clove oil, has been used in dentistry for decades as an analgesic and has demonstrated excellent biocompatibility[42]. With an acceptable dietary intake upper value of 2.5 mg kg$^{-1}$ day$^{-1}$, eugenol can be efficiently excreted by liver[43] and the lethal dosage (LD$_{50}$) is reported to be 11 mg kg$^{-1}$ in rats[44]. Although there are few investigations on the toxicity of poly(eugenol), it has been used in biosensors with desirable biocompatibility[40,45]. The disintegration of ultrathin poly(eugenol) and possible degradation into eugenol could result in biocompatible products that can be metabolized by cells and organs[40,46].

As shown in Supplementary Fig. 1, PLLA–PTMC is dropcasted on customized frosted glass templates followed by sputtering of Au nanomembranes to achieve a large specific area. The frosted glass substrate with a high mesh size of 2000 is chosen to promote the detection sensitivity, as a rougher surface

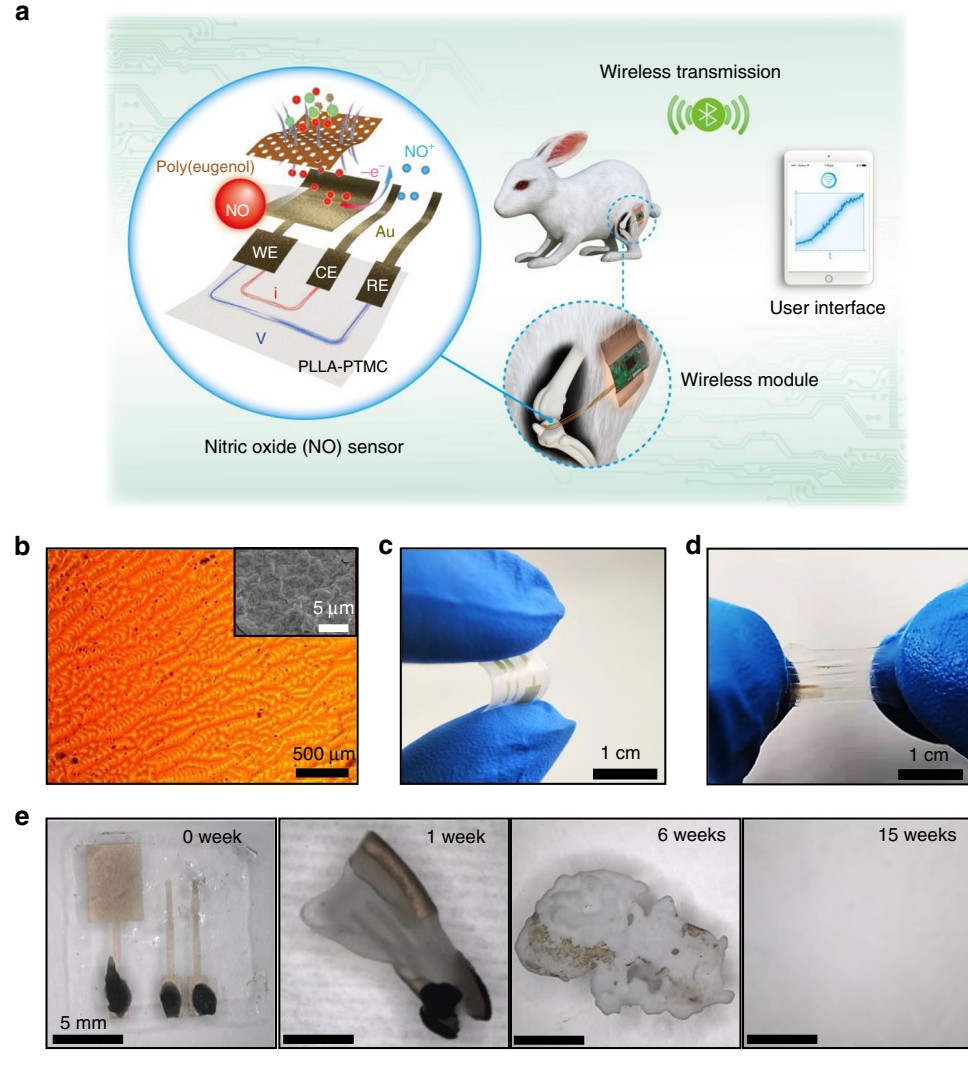

**Fig. 1 Materials and designs for flexible and transient nitric oxide (NO) sensors. a** Schematic illustration of a transient NO sensor composed of a bioresorbable PLLA–PTMC substrate (thickness: 400 μm), Au nanomembrane electrodes (thickness: ~32 nm), and a poly(eugenol) thin film (thickness: ~16 nm). NO concentration can be measured through amperometry by applying an oxidation potential between the working electrode (WE) and the reference electrode (RE) and measuring the current between the WE and the counter electrode (CE). The sensor implanted in the joint cavity of a New Zealand white rabbit can continuously detect NO concentrations in vivo and transmit the data to a user interface through a customized wireless module. **b** Optical image of the surface morphology of Au electrodes with poly(eugenol) films fabricated on frosted glass. Inset: scanning electron microscopy (SEM) image of the surface morphology. **c** Photograph of the NO sensor under bending. **d** Photograph of the NO sensor in a stretched state. **e** Images collected at various stages (0, 1, 6, and 15 weeks) of accelerated degradation of a transient NO sensor in phosphate-buffered saline (PBS) solutions at 65 °C.

yields a greater response current to NO (Supplementary Fig. 2). The surface morphology of the Au nanomembrane appears in Fig. 1b and Supplementary Fig. 3. The tacky surface of PLLA–PTMC assures the excellent adhesion of the Au nanomembrane without using additional adhesive layers. Poly (eugenol) is electrochemically deposited on the surface of the WE to minimize sensing interferences induced by associated chemicals in the biological systems, such as glucose, nitrites, uric acid, etc. For electrochemical deposition, higher concentrations of eugenol in the electrolyte provide better anti-interference performance, but result in lower current responses to NO (Supplementary Fig. 4). For optimal performance, a eugenol concentration of 10 mM is chosen to tradeoff between the anti-interference and the NO sensitivity. The functional groups of the deposited poly(eugenol) film on the WE are characterized by Fourier transform infrared spectrometry (Supplementary Fig. 5). Compared with bare Au on the PLLA–PTMC film, the poly

(eugenol) coating shows a combination of aliphatic and aromatic characters with different oxygen-containing groups, which is consistent with previous studies[47]. The height profile measured by a profilometer shows that the thickness of the deposited poly (eugenol) is approximately 16 nm (Supplementary Fig. 6). A biodegradable paste made from a mixture of PLLA–PTMC and molybdenum (Mo) particles serves as the electrical connector to the testing wires. Previous studies have revealed that Mo is biodegradable in aqueous environments[48] and the recommended dietary allowance for adults is 45 μg day$^{-1}$[49]. Mo has been used as dissolvable electrodes and interconnects for various transient electronics, such as biodegradable batteries, intracranial pressure sensors, and neural sensors[21,22,50]. The particle size and proportion of Mo are optimized to achieve a high electrical conductivity (resistance below 10 Ω) (Supplementary Fig. 7). The combination of the bioresorbable and highly stretchable substrate, ultrathin electrodes and selective membranes gives the sensor the

desirable flexibility and stretchability, as shown in Fig. 1c, d. The resistance of the sensor electrode remains unchanged after 1000 cycles of tensile tests at strains of 20 and 50% (Supplementary Fig. 8a), and 1000 cycles of bending tests at angle up to 90° (Supplementary Fig. 8b). It is noted that the resistance of Au electrodes increases over different stains upon stretching (Supplementary Fig. 8c), which could affect the oxidation potential and response current for NO detection (Supplementary Fig. 8d). It is therefore important to minimize the strains of Au electrodes during the course of NO sensing.

An accelerated soaking test in phosphate-buffered saline (PBS) at 65 °C shows that the NO sensor is capable of complete physical transience after 15 weeks, based on the degradation process given in Fig. 1e and Supplementary Fig. 9. The hydrolysis of the PLLA–PTMC substrate induces swelling of the substrate, followed by disintegration of the Au and poly(eugenol) nanomembranes and the dissolution of Mo paste, resulting in the eventual disappearance of the entire device.

**Device characterization**. The detection of NO is based on amperometry using a standard three-electrodes configuration, with Au nanomembranes serving as the WE, CE, and RE. The redox reaction involves the oxidation of a molecule of NO with one unpaired electron to $NO^+$ (nitrosonium ion) on the surface of the WE, followed by a subsequent conversion to $NO_2^-$ in the solution[9]:

$$NO - e^- \rightarrow NO^+. \tag{1}$$

$$NO^+ + OH^- \rightarrow HNO_2. \tag{2}$$

The redox current can therefore be monitored to detect the concentration of NO. The performance of fabricated NO sensors is evaluated at 37 °C. As shown in Fig. 2a, an oxidation potential of approximately 0.8 V (WE vs. RE) are determined through linear sweep voltammetry (LSV) by measuring the response current between the WE and CE in the presence of NO. The time dependent current response at different NO concentrations are measured through chronoamperometry and the results are shown in Fig. 2b, c. During the course of the measurement, stirring is required upon each addition of NO standard solutions to achieve uniformity, followed by data recording in the absence of stirring to minimize noises and ensure data stability especially at low NO concentrations (<1 μM, response current < 11 nA). As the NO concentration increases, an increase in the current response can be rapidly captured (<350 ms), which is important for real-time NO monitoring. The subsequent current attenuation is mainly attributed to the relatively sluggish diffusion of NO to the electrode surface in the PBS. As shown in Fig. 2d, e, a linear relationship between the NO concentration and the response current can be obtained (calibration curves), and the detection sensitivities are calculated to be 5.29 and 4.17 nA μM$^{-1}$ for NO concentrations in the range of 0–5 and 5–100 μM, respectively. Based on the calibration curve, the detection limit of the sensor is 3.97 nM, with the calculation details given in the "Methods" section.

The selectivity and specificity of the NO sensor are investigated using common interfering substances in biological systems including glucose (GLU), sodium nitrite (Nitrite), sodium nitrate (Nitrate), ascorbic acid (AA) and uric acid (UA) and the experimental results appear in Fig. 2f. Given the additions of high concentrations of NO and interfering chemicals, stirring can be continuously applied to achieve uniformity and yet maintain stable response current. The anti-interference performance is evaluated by determining the ratio of the response current toward the interfering chemicals to that toward NO (Fig. 2g). Data are shown as the means ± standard deviations with $n = 3$

independent experiments. With the addition of potential interfering chemicals at concentrations of 5 times (0.5 mM) that of NO (0.1 mM), the device obtains the strongest signal to NO and the response current ratio of interfering chemicals is less than 15% (Fig. 2g). These results suggest that the sensor has excellent anti-interference characteristics, which are attributed to the hydrophobic property, ionic interaction and molecule size exclusion of the selective poly(eugenol) film. The uncharged small NO molecules can easily permeate through while large molecules (glucose, UA, AA, etc.) and negative ions ($NO_2^-$, $NO_3^-$, etc.) are blocked by the poly(eugenol) membrane[51]. Although molecules with positive charges (e.g., dopamine) or uncharged molecules (e.g., $H_2O_2$) could still permeate through poly(eugenol) membrane[51], $H_2O_2$ has limited influence due to the relatively low response current compared to that of NO[52] and dopamine are often present in a small amount. These species are not expected to have significant effects on NO measurements in most cases. Nevertheless, multi-functional selective membranes would need to be developed if exclusion of these chemicals is necessary.

Stability tests of the NO sensor demonstrate that the linear relationship between NO concentration and response current can be maintained up to 14 days, with the slope staying almost constant over 7 days and then gradually increasing to 1.5 times that of the initial state on the 14th day (Fig. 2h). The response current ratio of interfering chemicals to NO remains almost unchanged for GLU, nitrite and nitrate, and increases slightly for AA and UA over 7 days, while a dramatic increase is observed in all interfering chemicals except GLU on the 10th and 14th day; this is probably due to the gradual degradation of the poly (eugenol) film over time. Overall, the excellent stability of the sensor over 7 days is attributed to the stable Au nanomembrane electrode and the slow degradation rate of the poly(eugenol) film and PLLA–PTMC substrate. It is noted that the interfering chemicals are often present in physiological environments in lower concentrations compared to those used for the selectivity measurement, thus the desirable performance could be sustained to longer period beyond 7 days. Nevertheless, further improvement of the stability of NO sensors can be achieved by depositing thicker poly(eugenol) films, which could sacrifice the detection limit to a certain extent.

To investigate the biocompatibility of the NO sensor, the device is coincubated with human aortic vascular smooth muscle cells (HA-VSMCs) and the results appear in Fig. 3a, b and Supplementary Figs. 10 and 11. The merged fluorescent images of cell proliferation for 5 days (Fig. 3a) and the corresponding optical microscopy images (Supplementary Fig. 11) indicate no significant difference between the sensor group and the control group. Similar cell viability is also observed between the cells cultured on the NO sensor and in the control group, indicating an excellent biocompatibility of the constituent materials (Fig. 3b).

**Real-time NO monitoring of living cells and organs**. The real-time measurement of NO released from cells and organs is of great interest to study the correlation of NO generation with neuronal signaling and inflammatory responses. It is known that nitric oxide synthase (NOS) in chondrocytes can generate NO by the conversion of L-arginine (L-Arg)[53–55], and interleukin-1 beta (IL-1β) and N$^\omega$-nitro-L-arginine methyl ester (L-NAME), respectively function as NOS stimulators and inhibitors[56,57]. Shown in Fig. 3c, chondrocytes are seeded with IL-1β in PBS at 37 °C to simulate the condition of osteoarthritis. A surge in the response current is observed when L-Arg (5 mM) is added, and the current drops back to the baseline with the addition of L-NAME (10 mM) that inhibits NO production. In comparison, no

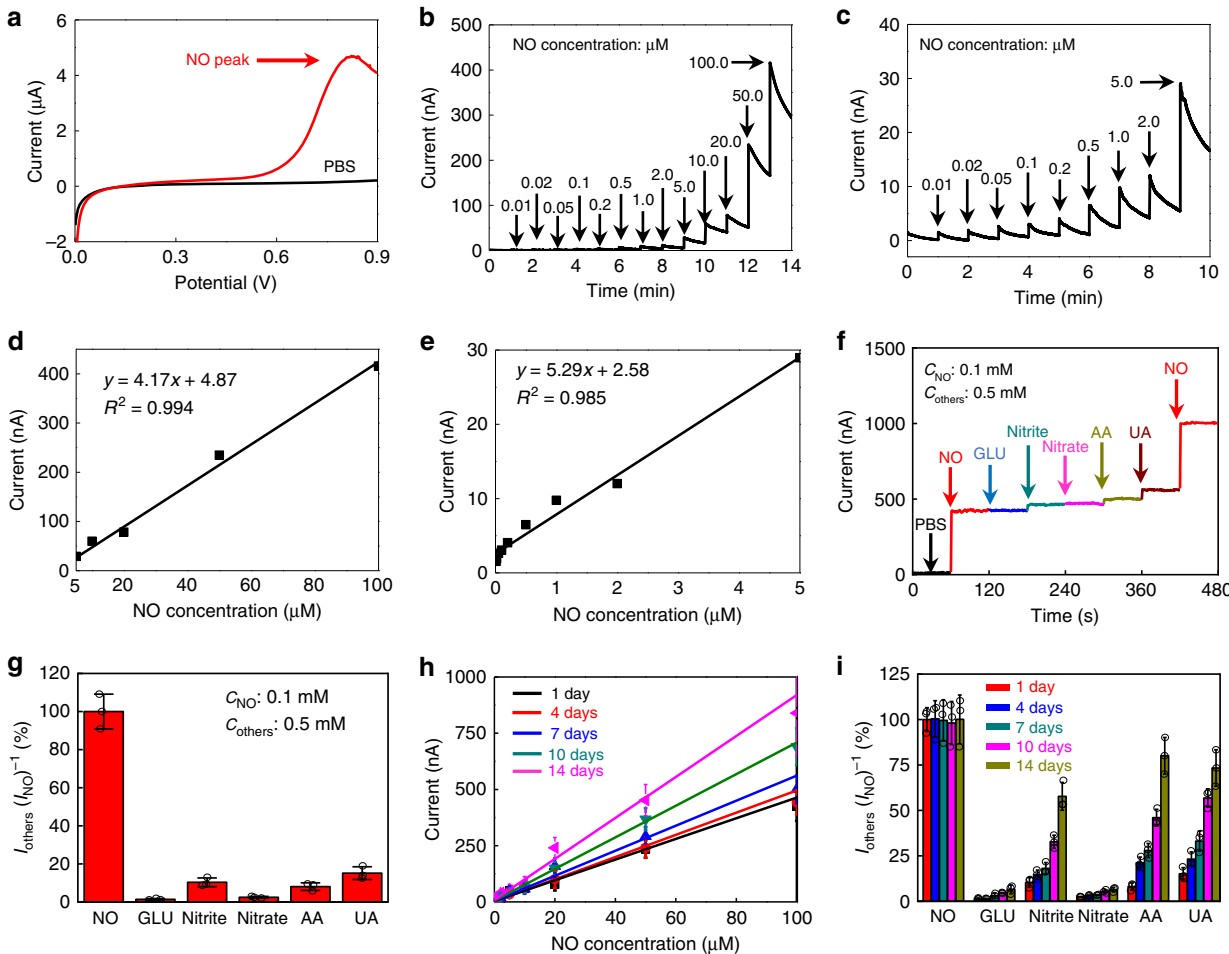

**Fig. 2 Electrochemical performance of a flexible and transient NO sensor at 37 °C. a** Linear sweep voltammetry performed in PBS solutions with (black) and without NO (red). **b** Time dependent current response of the sensor at different NO concentrations (detection range: 0–100 μM), at a bias voltage of 0.8 V. **c** Time dependent current response of the sensor at different NO concentrations (detection range: 0–5 μM) at a bias voltage of 0.8 V. **d** Calibration curve: linear relationship between response currents and NO concentrations (5–100 μM). **e** Calibration curve: linear relationship between response currents and NO concentrations (0–5 μM). **f** Selectivity measurement: current response with the additions of a series of potential interfering chemicals (0.5 mM) and NO solutions (0.1 mM) (GLU, glucose; Nitrite, sodium nitrite; Nitrate, sodium nitrate; AA, ascorbic acid; UA, uric acid). **g** Quantitative analysis of the selectivity of the NO sensor. Signal is defined by $I_{others}(I_{NO})^{-1}$, where $I_{others}$ is the response current with the addition of interfering chemicals (0.5 mM) and $I_{NO}$ is the response current with the addition of the NO solution (0.1 mM). **h** Stability of the NO sensor: linear relationship between response currents and NO concentrations from 1 to 14 days. **i** Stability of the NO sensor: selectivity measurement from 1 to 14 days. In **a–i**, $n = 3$ independent experiments. In **g–i**, data are shown as the means ± standard deviations.

response current is detected in PBS in the absence of chondrocytes with the same additions of L-Arg and L-NAME, indicating that they are not interfering with NO detection. The data of interfering tests of L-Arg and L-NAME toward NO detection are given in Supplementary Fig. 12a. These results suggest that a real-time NO variation in chondrocytes can be captured by the sensor. The tracking of NO concentration in chondrocytes over a 24-h period by the NO sensor is presented in Fig. 3d. The concentration of NO is converted based on the calibration curve in Fig. 2e. Continuous NO generation in chondrocytes is detected with the additions of L-Arg and IL-1β after the sixth hour. As a comparison, a standard Griess test for NO detection based on the measurement of nitrites in the solution is also employed to investigate NO release by sampling the solutions at different stages (every 2 h) and analyzing the $NO_2^-$ concentration afterwards. The calibration curve for the Griess method is shown in Supplementary Fig. 12b. It should be noted that the Griess method only provides the accumulative amounts of $NO_2^-$, thus indirectly measuring NO in the solution. Both the results from the sensor and Griess tests demonstrate a similar trend in NO

generation (Fig. 3d). These results suggest that real-time NO concentrations in chondrocytes are successfully captured by the sensor, which can offer a dynamic monitoring of NO release that cannot be achieved with the standard Griess test.

The real-time monitoring of NO release from living organs of mammals is also performed using the NO sensor in vitro. Shown in Fig. 4a, in PBS solutions with the presence of livers of Sprague Dawley (SD) rats, an increase in the response current is detected after L-Arg (5 mM) is added. The response signal declines when L-NAME (1 mM) is added simultaneously with L-Arg, and is completely suppressed when the concentration of added L-NAME is increased to 5 mM. By contrast, there is no current response in the solutions without the liver after adding L-Arg or L-Arg with L-NAME. These results indicate that organ activity of NO release is successfully detected by the sensor. A similar increase of NO signals is recorded after the addition of L-Arg in PBS solutions with rat brain, kidney, and heart (Fig. 4b), as well as various organs of a New Zealand rabbit (Fig. 4c–f). Although the same concentration of L-Arg (5 mM) is added to promote NO release, response signals appear different in various organs, probably due

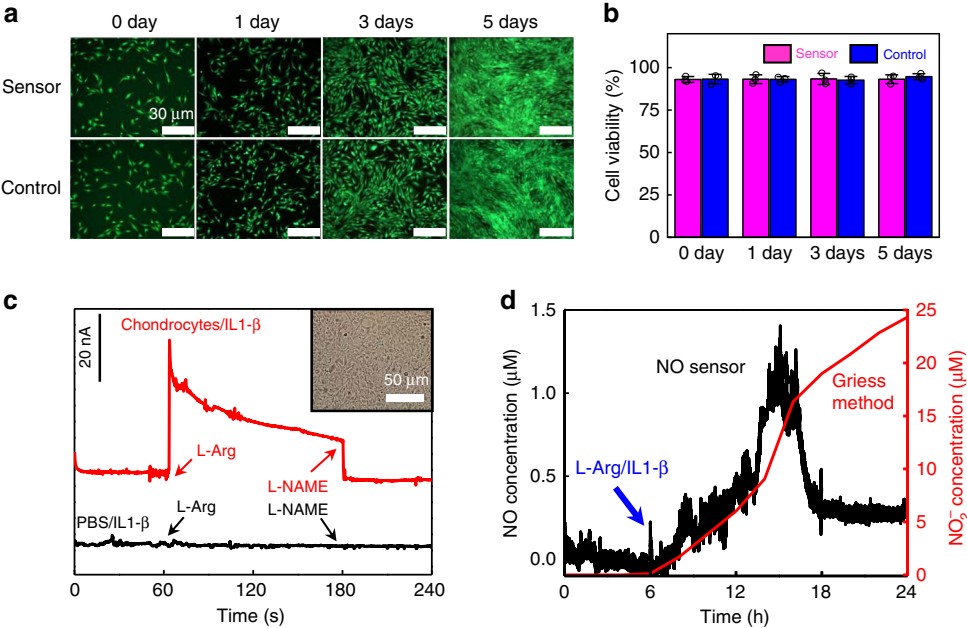

**Fig. 3 Cell cytotoxicity tests of NO sensors and real-time NO detection in chondrocytes at 37 °C. a** Fluorescent images of human aortic vascular smooth muscle cells (HA-VSMCs) cultured on NO sensors with Calcein-AM/Propidium Iodide (Calcein-AM/PI) staining. Green (Calcein-AM) for live cells and red (PI) for dead cells. **b** Cell viability over 0, 1, 3, and 5 days. **c** Real-time current response of NO sensors, with the addition of L-arginine (L-Arg) to promote NO release and $N^\omega$-nitro-L-arginine methyl ester (L-NAME) to inhibit NO release. Red: rat chondrocytes cultured in PBS with interleukin-1 beta (IL1-β); black: PBS with IL1-β. Inset: optical image of chondrocytes. **d** Real-time measurement of NO concentrations over 24 h in chondrocytes cultured in PBS by the NO sensor (black), in comparison to accumulated $NO_2^-$ concentrations measured by standard Griess tests (red). L-Arg and IL1-β are added to promote NO release. In **a–d**, $n = 3$ independent experiments. In **b**, data are shown as the means ± standard deviations.

to variant amounts of NOS and different diffusion kinetics of NO in these organs. The continuous detection capability of the sensor for NO signals can play an essential role in probing the functional pathways of NO in many organs.

**Real-time in vivo NO monitoring in living mammals.** Although the NO detection in biological environments has long been considered to be challenging, the NO sensor developed in the current work offers the potential of real-time NO monitoring in vivo, due to its desirable sensitivity, selectivity, and stability (≥7 days). Herein, we demonstrate two scenarios of continuous in vivo NO monitoring to provide essential information for critical biological events (Fig. 5).

Nitroglycerine (NTG) is known to be a medicine to treat angina pectoris, myocardial infarction and chronic heart failure by releasing NO to dilate the vascular system[58,59]. Proper dosages of NTG are critical because the correct dose varies among different patients and overdosages can result in reflex tachycardia[60]. Therefore, the real-time monitoring of NO release in the cardiovascular system can provide essential feedback to promptly adjust the NTG usage. In the current experiment, NTG injection is transfused via the ear–vein of New Zealand white rabbits with a controlled speed by an infusion pump and the NO release is determined by a sensor placed between the pericardium and the beating heart. Shown in Fig. 5a, the response current starts to increase immediately after the NTG is transfused, indicating the elevated generation of NO in the heart region. Interestingly, the NO sensor can simultaneously record the electrocardiography (ECG) signals upon NO detection, as shown in Fig. 5a and Supplementary Movie 1, which can be used to identify possible arrhythmia and help adjust the NTG dose. As a comparison, the fluids near the heart are simultaneously sampled at different times (every 2 min) to obtain the NO concentration

by the Griess assay. The detection results of the NO sensor align well with those measured by the standard Griess method (Fig. 5b), validating the efficacy of the NO sensor. The flexible and transient NO sensor enables unique continuous monitoring of NO and ECG in the cardiovascular region and is important in providing effective and timely therapeutic treatments and pharmaceutical guidelines.

On the other hand, as chondrocyte apoptosis and cartilage degradation are closely related to NO concentrations, detection of NO in the joint cavity can provide essential diagnostic information for treatment protocol and postsurgical monitoring of osteoarthritis. Current investigations of NO content released from chondrocytes are performed by joint fluid extraction followed by chemiluminescence, fluorescence, biofluid, or Griess assay, which preclude the NO detection with high spatial and temporal resolutions[6,61–64]. By contrast, the flexible and transient NO sensor can dynamically monitor NO levels in a specific location. To demonstrate its utility, NO sensors are implanted into the joint cavity of New Zealand white rabbits through surgical operation to monitor NO signals over time (Fig. 5c, Supplementary Fig. 13). In particular, a battery powered, wirelessly operated circuit module is designed to enable NO signal readout (Supplementary Movie 2). Moreover, a customized software package is developed to wirelessly transmitted the data via a Bluetooth connection and display them in a mobile device. The schematic diagram of the wireless module appears in Fig. 5d. The wireless circuit is immobilized on the thighs of the rabbits by surgical tape and is connected to the NO sensor implanted transcutaneously. The implantation site and the location of the wireless module are shown in Fig. 5e, f, and the circuit design and system architecture appear in Supplementary Figs. 14 and 15. It is noted that the nondegradable wireless module at this stage can demonstrate advanced remote diagnostic functions of NO sensors. To achieve an entirely transient sensing platform, future

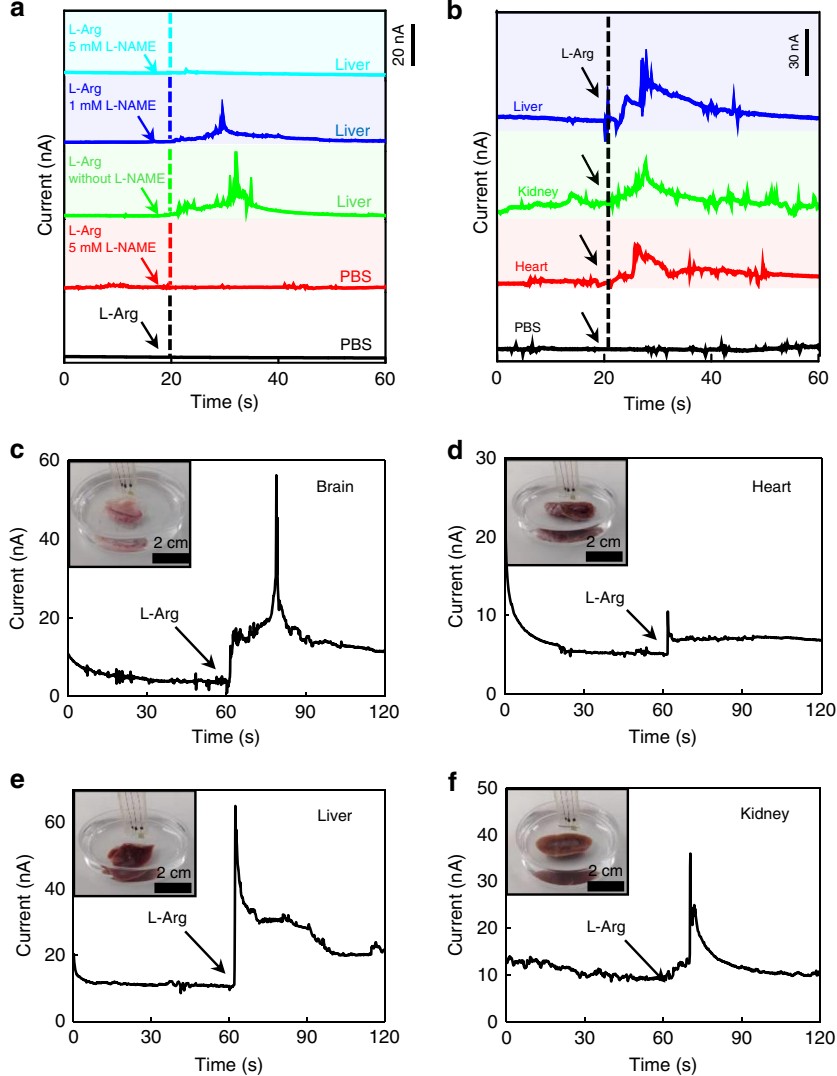

**Fig. 4 Ex vivo real-time detection of NO generated from different organs. a** Time dependent current response of NO from the rat liver, in the addition of ʟ-Arg and different concentrations of NO enzyme inhibitor ʟ-NAME. **b** Time dependent current response of NO from rat liver, kidney, and heart, with ʟ-Arg to promote NO release. **c–f** Time dependent current response of NO from different organs of a rabbit. **c** Brain; **d** heart; **e** liver; **f** kidney. Insets show photographs of NO detection of living organs in PBS. ʟ-Arg is added to promote NO release. In **a–f**, $n = 3$ independent experiments.

efforts are needed to develop degradable data transmitting systems by leveraging established CMOS foundry techniques, the potential feasibility of which has been demonstrated by previous works[65,66]. The investigation of NO concentrations in the joint cavity over a 5-day period is performed under three conditions, including the control (sensor implantation without treatment), the penicillin (antibiotic treatment after sensor implantation) and the IL-1β (promoting inflammation after sensor implantation) groups. The NO signal is monitored for 1 h every day and the recorded response currents are given in Fig. 5g, h and Supplementary Fig. 16. The corresponding NO concentrations converted from the calibration curve (Fig. 5d, e) over the 5-day period are summarized in Fig. 5i. One can observe that the NO concentration of the IL-1β group is significantly higher than that in the other two groups, while there is no obvious difference between the control group and the penicillin group. The above results indicate that inflammation in the joint cavity of the rabbit is associated with a high concentration of NO, which is consistent with the results from the previous reports[67,68]. The stable sensing characteristics of the NO sensor in the joint cavity region and its capability of establishing a link between NO concentrations and

inflammation responses could offer essential information to optimize osteoarthritis treatments.

Moreover, despite the residual copper connection wires to the wireless module and the nondegradable sutures for implantation, the implanted NO sensor completely disappears after 8 weeks of implantation in the articular cavity, as shown in Supplementary Fig. 17. Hematoxylin–eosin (HE) staining of tissues at the implantation sites shows no obvious inflammation signs or any residuals of the PLLA–PTMC substrate and Au nanomembrane electrodes (Fig. 5j, Supplementary Fig. 18). Further evaluation of the element content of Mo and Au in the surrounding tissues of the sensor, liver, kidney and urine through inductively coupled plasma mass spectrometry (ICP–MS) suggest no detectable accumulation compared to that of the control group (Supplementary Fig. 19). These results suggest that the NO device built with ultra-thin Au (~12.4 μg) and poly(eugenol) (~0.3 μg) layers on biodegradable PLLA–PTMC substrates is capable of full physical transience in vivo after 8 weeks of implantation, through hydrolysis of PLLA–PTMC, disintegration of Au and poly (eugenol) nanomembranes and eventual potential clearance through phagocytosis and renal metabolization.

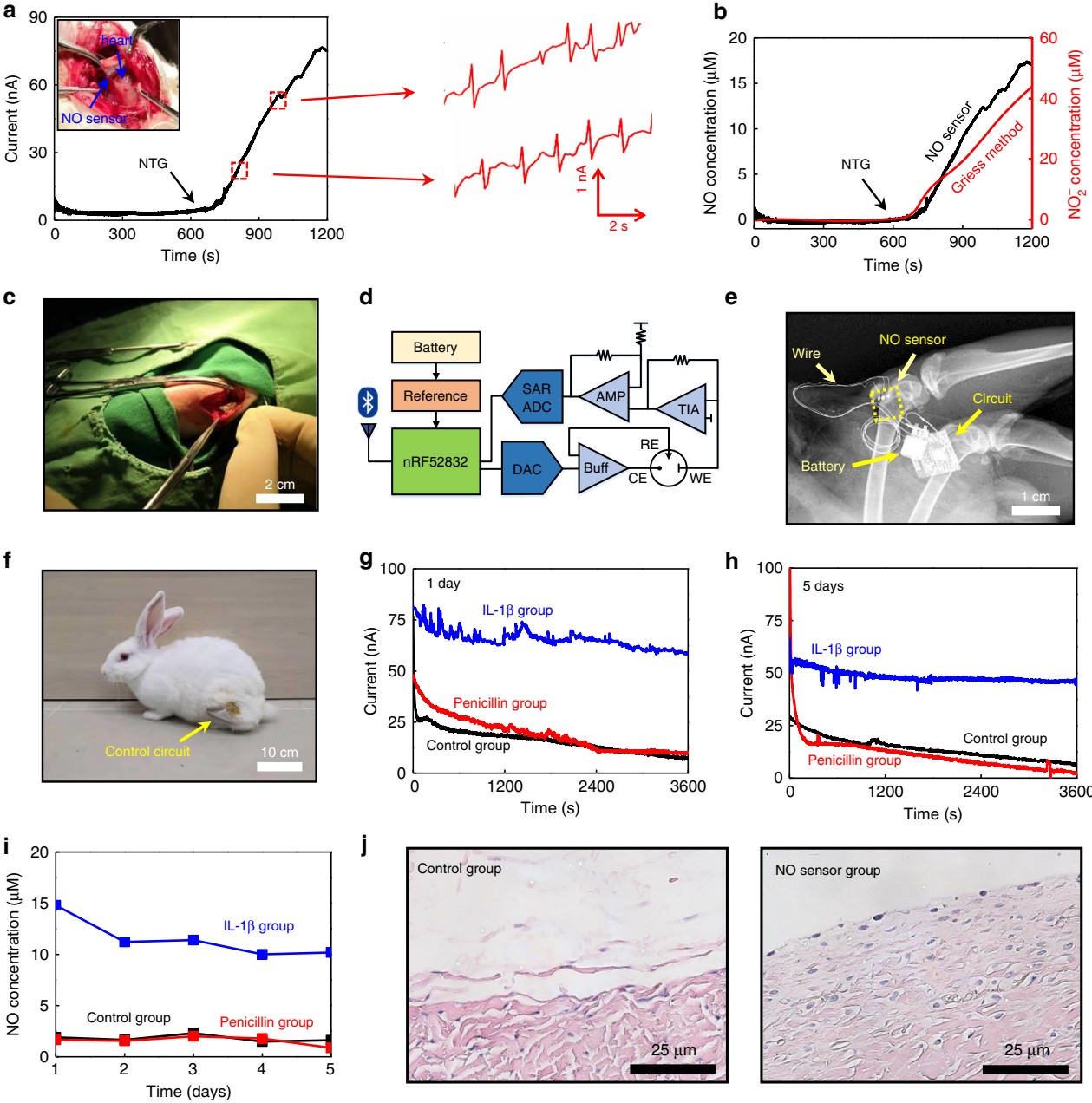

**Fig. 5 In vivo real-time monitoring of NO concentrations in the heart and joint cavity of New Zealand rabbits. a** Real-time measurement of the response current of the NO release from the heart of the rabbit, stimulated by the intravenous infusion of nitroglycerine (NTG), with simultaneous electrocardiography (ECG) recordings from the NO sensor in the enlarged view. **b** Real-time monitoring of the NO concentration in the heart of a rabbit stimulated by NTG infusion (black), in comparison to accumulated $NO_2^-$ concentrations measured by standard Griess tests (red). **c** Photograph of the surgical implantation of the NO sensor in the joint cavity of a rabbit. **d** Schematic diagram of wireless control and transmission system for the NO sensor. SAR ADC, successive approximation register analog-to-digital converter; DAC, digital-to-analog converter; Buff, buffer; AMP, amplifier; TIA, transimpedance amplifier. **e** X-ray image of the implanted NO sensor, wire connections and wireless module. **f** Photograph of the rabbit after NO sensor implantation with the wireless circuit immobilized on the thigh. **g** Real-time monitoring of the current response of NO after 1 day of implantation of the NO sensor in the joint cavity of a rabbit. **h** Real-time monitoring of current response after 5 days of implantation of the NO sensor in the joint cavity of a rabbit. **i** Real-time monitoring of NO concentration over 5 days (NO concentrations are converted from the measured response currents). In **g–i**, measurements are performed in three groups of rabbits after sensor implantation: IL-1β treatment (blue); penicillin treatment (red); and control group with no treatment (black). **j** Hematoxylin–eosin (HE) staining images of tissues at the implantation site after 8 weeks. In **a–c** and **g–j**, n = 3 independent experiments.

## Discussion

We report a flexible and physically transient electrochemical NO sensor with a wide sensing range (0.01–100 μM), a low detection limit (3.97 nM), a fast response time (<350 ms), desirable stability (≥7 days), and excellent anti-interference characteristics compared with those of previously reported electrochemical NO sensors (Supplementary Table 1). The materials strategies and device designs of the NO sensor enable unique flexible and degradable characteristics and the competence of continuous monitoring in physiological environments. The entire NO device is capable of full degradation after 8 weeks of implantation in vivo without introducing apparent inflammation or toxicity. The

flexible and transient features of the sensing platform enable implantation with minimal irritation and avoid additional surgical procedures for device retrieval. Real-time monitoring of NO concentration is realized both in vitro (chondrocytes and major organs of rats and rabbits) and in vivo (heart and join cavity regions) with wireless control and data transmission capability. Future directions include the development of biodegradable encapsulation materials to ensure durable electrical contacts to the wireless modules, and fully implantable and transient control circuits that can be wirelessly powered with biodegradable antennas and/or batteries. In addition, it is envisioned that miniaturized sensor arrays can be implemented to spatially resolve NO distributions in different body regions. Choosing proper selective membranes and enzymes, electrochemical sensors that can probe other important biomarkers (dopamine, glucose, etc.) can be fabricated based on similar device designs. Considering the diverse demands in biomedicine, the NO sensors could be integrated with other devices like electrical stimulators and microfluidic channels, to realize close-loop, multifunctional physiological monitoring, and interrogation. Collectively, the device strategy may potentially offer unique approaches to study neuroscience and disease pathology, and provide essential therapeutic and diagnostic information to evaluate immune/inflammatory responses, establish pharmaceutical guidelines, optimize treatment protocols, and realize continuous postsurgical monitoring that forms the essential baselines to improve healthcare.

## Methods

**Fabrication of the NO sensor**. PLLA–PTMC (30:70) with a viscosity of 2.1 mpa.s (Jinan Daigang Biomaterial Co., Ltd., China) was dissolved into trichloromethane (CHCl$_3$, Beijing Tongguang Chemical Co., Ltd., China) with a weight to volume ratio of 1:10, followed by drop-casting of the solution on a surface of customized frosted glass (Guangzhou Hongxing Chemical Co., Ltd., China) with a mesh size of 0, 1000, and 2000. The PLLA–PTMC films were cured for 12 h at 4 °C to avoid bubble formation. The films were then peeled off the substrate for Au nanomembrane (~32 nm) deposition in a magnetron sputter (Beijing Zhongjingkeyi Co., Ltd., China) with a deposition speed of 0.7 Å s$^{-1}$ (380 V, 0.06 A) through patterned shadow masks. Selective poly(eugenol) membranes were electrochemically deposited on the WE. A deaerated NaOH (60 mL, 0.1 M, Beijing Tongguang Chemical Co., Ltd., China) solution with additions of eugenol (5, 10, and 15 mM, Annaiji Chemical Co., Ltd., China) served as an electrolyte. For deposition, Pt and Ag/AgCl electrodes were used as CE and RE respectively. Cyclic voltammetry with a scan speed of 20 mV s$^{-1}$ was performed from 0 to 0.7 V with the WE for 10 cycles to achieve poly(eugenol) deposition and was followed by rinsing the sample with deionized water to remove residual electrolytes. The Au electrodes were connected to the copper wires for external testing and wireless communication, through a biodegradable conductive paste made of PLLA–PTMC and Mo particles. The conductivity of the paste was investigated with different Mo particle sizes and weight ratios. The connection area was encapsulated by 3140 adhesive (Dow Coming Corp. USA) and poly(dimethyl siloxane) (PDMS) to ensure stable electrical contact for NO detection.

**Wireless control and data transmission system**. Key modules, including the analog front-end, digital control logic and power management, have been designed based on off-the-shelf components. A transimpedance amplifier, based on an optional amplifier, converting collected current to voltage that can be used for processing and a voltage amplifier has been included in the analog front-end. The control logic, regulating the analog front-end with the digital-to-analog converter and the successive approximation register analog-to-digital converter has been implemented in a microprogrammed control unit (MCU). An iOS-based software program dedicated to NO detection has been developed to realize wireless transmission based on a Bluetooth module in the MCU. The power management includes Li-battery charging and regulator circuits, based on a low dropout regulator and reference for voltage. The entire power consumption of the circuit is less than 20 mW. The proposed system features a volume size of $2.23 \times 1.76 \times 0.83$ cm, with a weight of 1.2 g.

**Preparation of standard NO solution**. NO gas was produced by a dropwise addition of 3 M sulfuric acid solution into a saturated nitrite sodium solution, and was purified by bubbling the gas through 1 M NaOH solution twice. A saturated NO solution (1.7 mM, 37 °C) was prepared by bubbling the generated NO gas into PBS solutions. The saturated solution was then diluted with PBS to obtain different NO concentrations to establish a calibration curve for a NO sensor. The NO

solutions were freshly prepared for each experiment to ensure reliable NO concentrations.

**Electrochemical NO measurement**. Electrochemical tests were performed on a CHI 650E electrochemical work station (Shanghai Chenhua Co., Ltd., China) at 37 °C. For NO oxidation potential determination, LSV was adopted with a scan rate of 20 mV s$^{-1}$ from 0 to 0.9 V. An amperometry method was employed for both in vitro and in vivo NO detection using the oxidation potential obtained from LSV. Before each test, the electrode system was stabilized for 1 h at the NO oxidation potential. NO detection was conducted in a Faraday cage to avoid electromagnetic disturbance from the surrounding environment. To acquire an accurate and stable response current signal for the NO calibration curve, especially at low concentrations, mechanical stirring was applied to achieve a uniform NO concentration before recording the response current, and then stirring was turned off during the short period of data recording. The detection limit of the sensor can be calculated from the calibration curve by $3S_b m^{-1}$ where $S_b$ is the standard deviation of the intercept of the calibration curve and $m$ is the slope of the calibration curve[69]. For selectivity tests, NO and interfering chemicals (glucose, sodium nitrite, sodium nitrate, ascorbic acid, and uric acid) were added in sequence in PBS, and the response current was recorded with mechanical stirring. The concentrations for NO and interference chemicals were 0.1 and 0.5 mM, respectively. Interfering tests were also performed on L-Arg (5 mM) and L-NAME (10 mM). The selectivity of the sensor can be evaluated by the ratio of the current response of different interfering chemicals to the current response of NO. Continuous stirring can be applied throughout the measurement due to the relatively high concentrations of NO and interfering chemicals. For the stability test, the sensor was immersed in PBS at 37 °C, and the calibration curves and sensor selectivity were obtained on the 1st, 4th, 7th, 10th, and 14th days.

**Cell cytotoxicity tests**. Cell cytotoxicity tests was conducted using a CCK-8 assay and Calcein-AM/Propidium Iodide (Calcein-AM/PI) staining. The HA-VSMCs (ATCC, CRL-1999, Manassas, VA, USA) were cultured in an RPMI-1640 medium supplemented with 10% fetal bovine serum (FBS) and penicillin (100 U mL$^{-1}$)/streptomycin (100 μg mL$^{-1}$). First, the sensor was sterilized by UV light 3 times for 30 min and then put into a 24-well plate with a cell density of $1 \times 10^5$ cells per well. The cells were incubated in 5% CO$_2$ at 37 °C. After incubation for 0, 24, 72, and 120 h, the medium was removed and 100 μL of CCK-8 reagents were added to each well to determine cell viabilities. A microplate reader (PerkinElmer, Waltham, MA, USA) was used to measure the optical density (OD) value at a wavelength of 450 nm. Meanwhile, after the CCK-8 reagents were removed, the cells were washed twice with PBS and then stained with Calcein-AM/PI (Biyuntian Co., Ltd., China). The fluorescence images were obtained with fluorescence microscopy (Leica Microsystems Inc., Buffalo Grove, IL, USA).

**In vitro NO detection in cells**. Chondrocytes from cartilage of SD rats (7 days) were cultured in DMEM/F12 medium with 10% FBS and 1% penicillin/streptomycin (all supplements were purchased from Gibco, Gaithersburg, MD, USA) at 37 °C with 5% CO$_2$. The medium was changed after 3 days. After reaching 70–80% confluency, chondrocytes were trypsinized and subcultured at approximately a 1:3 split rate. The third or fourth passage were used for the following experiments. For short-term NO detection, chondrocytes were put into 5 ml PBS with IL-1β (100 μL, 10 ng mL$^{-1}$) at 37 °C with a cell density of $1 \times 10^6$ cells per well. L-Arg (500 μL, 5 mM), as the NO reaction substrate, was added to promote NO release and L-NAME (500 μL, 5 mM), as the NO enzyme inhibitor, was added to inhibit NO generation. The corresponding response current was recorded. For the 24-h monitoring, the cells were seeded in plates and incubated with 10 mL of PBS at 37 °C, and IL-1β (200 μL, 10 ng mL$^{-1}$) and Arg (1 mL, 5 mM) were added at the 6th hour to promote NO generation. The solution in the well was extracted every 2 h for the Griess test. Concentrations of nitrite were determined by a Griess reagent kit (Thermo Fisher Scientific, Waltham, MA, USA), in which the sampled solutions and Griess reagent were added into each well and incubated for 30 mins at room temperature, followed by spectrophotometric measurement of the absorbance of each sample at 562 nm. The NO concentration measured from the sensor and Griess methods can be acquired according to their calibration curves (Fig. 2e, Supplementary Fig. 12).

**Ex vivo NO detection in organs**. SD rats and New Zealand white rabbits were sacrificed and the brain, heart, liver and kidney were partially removed for NO detection. The removed organs were placed in PBS solutions at 37 °C for NO detection. An amperometry method was performed to detect NO released from the cells and organs. L-Arg was used as the NO substrate, and L-NAME was used as the NOS inhibitor.

**In vivo NO detection in heart and joint cavity**. All animal procedures were completed in agreement with the institutional guidelines of the Beijing Institute of Traumatology and Orthopaedics. The experimental protocol was reviewed and approved by the Institutional Animal Care and Use Committee (IACUC) at Beijing Institute of Traumatology and Orthopaedics. For NO detection in the heart region, New Zealand white rabbits were anesthetized by Nembutal, and then the chest was

opened and fixed with a hemostat. The NO sensor was inserted between the beating heart and pericarditis. A glucose solution was first transfused via the ear vein followed by the infusion of NTG at a rate of 60 µg min$^{-1}$ controlled by a peristaltic pump. The response current was recorded throughout the process and converted to NO concentration based on the calibration curve of Fig. 2d, e. The liquid within the pericarditis was simultaneously sampled every 2 min for the Griess test. For NO detection in the joint cavity, the sensor was implanted in the joint cavity region in New Zealand white rabbits through a surgical operation. Three groups of rabbits with different treatments were investigated: (1) the IL-1β group: IL-1β treatment (2 mL, 20 ng ml$^{-1}$, one injection) after sensor implantation; (2) the penicillin group: penicillin treatment (8 × 10$^6$ units, one injection) after sensor implantation; and (3) the control group: no treatment after sensor implantation. After the NO sensor was implanted, the rabbit was anesthetized by Nembutal and the current response for NO was wirelessly recorded for 1 h every day for a 5-day period. The recorded data at 2400 s of the 1-h monitoring were chosen to represent the NO concentration level for each day. For HE staining, the tissues around the implantation location were cut into small pieces and fixed in formalin for 1 week. The tissues were then embedded in paraffin wax and cut into 4-µm slices. Sections were incubated with hematoxylin and eosin at room temperature and analyzed under an optical microscope. For in vivo degradation tests, sensors were implanted into the joint cavity of New Zealand white rabbits. After 8 weeks the rabbit was sacrificed, and the tissues around the sensor were separated and observed. Moreover, the surrounding tissues of the implanted sensor, the liver, kidney, and urine of the rabbit were obtained for ICP–MS to evaluate the residual concentration of Mo and Au.

**Reporting summary**. Further information on research design is available in the Nature Research Reporting Summary linked to this article.

## Data availability
All data supporting the finding of this study are present in the article and the Supplementary Information files. All raw and processed data are available from the corresponding author on reasonable request.

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

## Acknowledgements

This project was supported by the National Natural Science Foundation of China (51601103), Tsinghua University-Peking Union Medical College Hospital Initiative Scientific Research Program (20191080592), the China Postdoctoral Science Foundation (2017M620769), and Beijing Municipal Health Commission (BMC2018-4).

## Author contributions

R.L., H.Q., X.S., and L.Y. conceived and designed the research project. R.L., Y.D., S.L., L.W., C.H., X.S., and L.Y. designed and fabricated the devices and performed the analysis. R.L., J.H., and X.Z. performed the cell toxicity tests. R.L., Y.D., S.L., H.Q., Y.J., and J.J. performed the animal studies. R.L., Y.M., and M.Z. designed and fabricated the wireless control and transmission system. R.L., X.S., and L.Y. wrote the paper with input from all authors. R.L., H.Q., Y.M., and Y.D. contributed equally to the paper.

## Competing interests

The authors declare no competing interests.
