## [Peer Review File · Nature Communications]

Reviewers' Comments:

Reviewer #1:

Remarks to the Author:

This work presents the real-time nitric oxide (NO) monitoring using a flexible, biodegradable, and wireless electrochemical sensor with desirable anti-interference characteristics. The proposed sensor exhibits a low detection limit, a wide sensing range, a high response rate, and a superior selectivity based on biomaterials existed in the human body. The performance of the sensor was well evaluated through the cell/organ test in vitro and the animal experiments using the rabbit model in vivo. The test results show that the successful real-time NO monitoring in diverse environments with the good material biocompatibility. Furthermore, the sensor exhibits a comparable detection capability to the conventional colorimetric-based sensors, which proves its potential as a detector for the optimization of various clinical treatments and/or the post-surgical monitoring related to the release of the NO signal transmitter. The reviewer thinks that the novelty of the idea and the performance of the proposed device are sufficient to give enough inspiration to potential readers in the field of the unconventional biomedical electronics. Therefore, the reviewer recommends publication of this manuscript in Nature Communications after addressing some minor issues.

Comment #1: The explanation about the biodegradability related to gold and poly(eugenol), commonly regarded as non-biodegradable materials, seems insufficient. The authors described that the amount of such materials in the sensor is extremely small and may be disintegrated through phagocytosis and metabolization. However, the reviewer recommends to add more clear description about these materials in vivo. For example, the bio-inert property of the materials (although not bioresorbable or biodegradable) and the critical amounts that cause the potential liver-toxicity as well as related texts and references can be added to the revised manuscript.

Comment #2: The authors showed the high NO selectivity of the proposed sensor in comparison with many other chemicals in biological systems for demonstrating their superior anti-interference feature. This feature is attributed to the hydrophobic property, ionic interaction, and molecule size exclusion of the poly(eugenol) membrane. However, is there any chance that the small molecules with positively charged/uncharged properties can permeate such membrane? The related explanation for further clarification would be helpful.

Comment #3: The figure 2i indicates the stability and the selectivity of the NO sensor, and the selectivity dramatically decreases as time goes. Especially, within 14 days, the selectivity of the NO sensor to block the AA sensing seems to be low. This data may mean that the performance of the device lasts for about 1-2 weeks only. Further discussion can be added.

Comment #4: The figure 1d shows the stretchability of the sensor, however, the electrochemical performance of the stretched state is not described. It is recommended to add relevant data.

Reviewer #2:

Remarks to the Author:

This is an interesting paper dealing with a real time monitoring of nitric monoxide in living organism by an electrochemical sensor. In addition the parts of the sensor are claimed resorbable after certain time of being implanted in the mammal body. The originality level of the paper is high, however some important questions need to be addressed before it has been accepted for publication. These are listed below.

1) What is the rationale behind using a biodegradable electrochemical sensor if it is integrated with data transmitting electronic device which is not biodegradable. Also the sensor contains metals (Au, Mo) which are non-biodegradable. By the way, at the very end of the manuscript (Experimental Methods) the Authors state that the residual levels of these metals in body

tissues/liquids were evaluated by ICP-MS technique but the corresponding data is not reported. Please provide it.

2) There are some inconsistencies in the data presented on X-Y graphs. In particular, why the currents due to the addition of NO standards decay with time (Figs 2(b), (c)) while during interferences test they remain stable (Fig 2(f)). Additionally, L-arginine and L-NAME should be added to the list of tested interferents.

3) The list of references should be updated as follows:

Ref [41] has nothing to do with poly(eugenol). Please update accordingly.

Ref [46] is not adequate to the sequence it is listed in. It should be rather replaced by the very first paper reporting the application of poly(eugenol) as an NO-selective membrane i.e. Ciszewski A., Milczarek G. *Electroanalysis*, [10] 791 (1998).

Taking into account the above stated I suggest the acceptance of the manuscript for publication after minor revision.

Reviewer #1

Summary Recommendation: This work presents the real-time nitric oxide (NO) monitoring using a flexible, biodegradable, and wireless electrochemical sensor with desirable anti-interference characteristics. The proposed sensor exhibits a low detection limit, a wide sensing range, a high response rate, and a superior selectivity based on biomaterials existed in the human body. The performance of the sensor was well evaluated through the cell/organ test in vitro and the animal experiments using the rabbit model in vivo. The test results show that the successful real-time NO monitoring in diverse environments with the good material biocompatibility. Furthermore, the sensor exhibits a comparable detection capability to the conventional colorimetric-based sensors, which proves its potential as a detector for the optimization of various clinical treatments and/or the post-surgical monitoring related to the release of the NO signal transmitter. The reviewer thinks that the novelty of the idea and the performance of the proposed device are sufficient to give enough inspiration to potential readers in the field of the unconventional biomedical electronics. Therefore, the reviewer recommends publication of this manuscript in Nature Communications after addressing some minor issues.

Our response: We thank the reviewer for these favorable comments.

- **Comment #1:** “The explanation about the biodegradability related to gold and poly(eugenol), commonly regarded as non-biodegradable materials, seems insufficient. The authors described that the amount of such materials in the sensor is extremely small and may be disintegrated through phagocytosis and metabolization. However, the reviewer recommends to add more clear description about these materials in vivo. For example, the bio-inert property of the materials (although not bioresorbable or biodegradable) and the critical amounts that cause the potential liver-toxicity as well as related texts and references can be added to the revised manuscript.”

Our response: We thank the reviewer for the constructive suggestions. We have added more descriptions and relevant references on the biological properties of the gold and poly(eugenol).

Our modification to the manuscript: We added the following description on page 6-7:

“Biocompatibility and degradation studies of Au nanomaterials are mostly focused on nanoparticles, which have been proposed for various biomedical applications such as chemoradiation, photothermal therapy, drug delivery, et al. [45-47]. Despite of some contradictions, many reports suggest that gold nanoparticles are non-toxic with proper size and dosage, and metabolization occur through kidney, bladder or hepatobiliary system [48-52], e.g., no significant side effects were observed after 24 h with intravenous injection of gold nanoparticles ($\sim 100 \mu\text{g}/\text{rat}$) [53]. Although Au has been considered to be chemically inert, recent study reveals that gold nanoparticles (4-22 nm) are degraded in vitro by cells which is mediated by nicotinamide adenine dinucleotide phosphate (NADPH) oxidase in the lysosome, followed by a recrystallization process, indicating a potential metabolization mechanism for a trace amount of gold [54]. Poly(eugenol) layer (thickness $\sim 16 \text{ nm}$) is incorporated to promote sensing selectivity and specificity toward NO by hydrophobic repulsion, ionic interaction and molecular exclusion [55]. Eugenol, the main chemical component of clove oil, has been used in dentistry for decades as an analgesic and demonstrated excellent biocompatibility [56]. Eugenol with an acceptable dietary intake (ADI) upper value of $2.5 \text{ mg}/\text{kg}$ per day can be efficiently excreted by liver [57], and the lethal dosage LD_{50} of eugenol is reported to be $11 \text{ mg}/\text{kg}$ in male F-344 rats and $17 \text{ mg}/\text{kg}$ in male Syrian golden hamsters [58]. Although there are few investigations on toxicity of poly(eugenol), it has been used in biosensors with desirable biocompatibility [41, 59]. The disintegration of ultrathin poly(eugenol) and possible degradation into eugenol could result in biocompatible products that can be metabolized by cells and organs [41, 60].”

- ***Comment #2:*** *“The authors showed the high NO selectivity of the proposed sensor in comparison with many other chemicals in biological systems for demonstrating their superior anti-interference feature. This feature is attributed to the hydrophobic property, ionic interaction, and molecule size exclusion of the poly(eugenol) membrane. However, is there any chance that the small molecules with positively charged/uncharged properties can permeate*

such membrane? The related explanation for further clarification would be helpful.”

Our response: We thank the reviewer for this valuable comment. It is true that small interfering molecules with positively charged/uncharged properties could permeate poly(eugenol) membrane, such as H₂O₂ and dopamine. But H₂O₂ has limited influence on NO detection as the response current is much smaller compared to that of NO, and the concentration of dopamine is often very low in the body except the brain. These species will therefore not significantly affect NO measurements in most cases. Nevertheless, multi-functional selective membranes would need to be developed if exclusion of these chemicals are necessary. The discussions about small molecules with positively charged/uncharged properties are added in the manuscript.

Our modification to the manuscript: We added the following description on page 11:

“Although molecules with positive charges (e.g., dopamine) or uncharged molecules (e.g., H₂O₂) could still permeate through poly(eugenol) membrane [65], H₂O₂ has limited influence due to the relatively low response current compared to that of NO [66] and dopamine are often present in a small amount. These species are not expected to have significant effects on NO measurements in most cases. Nevertheless, multi-functional selective membranes would need to be developed if exclusion of these chemicals is necessary.”

- ***Comment #3:*** “The figure 2i indicates the stability and the selectivity of the NO sensor, and the selectivity dramatically decreases as time goes. Especially, within 14 days, the selectivity of the NO sensor to block the AA sensing seems to be low. This data may mean that the performance of the device lasts for about 1-2 weeks only. Further discussion can be added.”

Our response: We thank the reviewer for the suggestion. The selectivity decreases due to the gradual degradation of poly(eugenol) layer, indicating an operational time frames with excellent selectivity around 1-2 weeks. It has to be noted that these interfering chemicals are often present in physiological environments in lower

concentrations compared to those used for selectivity measurement, thus the desirable selectivity could still be sustained to longer period. Nevertheless, further improvement of the stability of NO sensors can be achieved by depositing thicker poly(eugenol) films, which could sacrifice the detection limit to a certain extent. We made further discussions about the selectivity and stability NO sensors.

Our modification to the manuscript: We added the following description to page 12:

“Overall, the superior stability of the sensor over 7 days is attributed to the stable Au nanomembrane electrode and the slow degradation rate of the poly(eugenol) film and PLLA-PTMC substrate. It has to be noted that the interfering chemicals are often present in physiological environments in lower concentrations compared to those used for selectivity measurement, thus the desirable selectivity could still be sustained to longer period. Nevertheless, further improvement of the stability of NO sensors can be achieved by depositing thicker poly(eugenol) films, which could sacrifice the detection limit to a certain extent.”

- **Comment #4:** *“The figure 1d shows the stretchability of the sensor, however, the electrochemical performance of the stretched state is not described. It is recommended to add relevant data”*

Our response: We thank the reviewer for the suggestion. We added 2 figures in Fig. S8 to demonstrate the characteristics of NO sensors. As the sensor stretches, the resistance of Au electrodes will change as a function of the strain (Fig. S8(c)). Different resistance of Au electrodes could affect the oxidation potential and response current for NO detection. The response current of Au electrodes with different resistance is shown in Fig. S8(d). It is therefore important to ensure that the resistance of Au electrodes does not change significantly during the course of NO detection. Fig. 1d and Fig. S8(a) are shown to indicate that the substrate of NO sensor is stretchable and after some cycles of stretching the resistance of Au electrodes return to the original state once the deformation is released. We added more discussions in the manuscript.

Our modification to the manuscript: We updated Fig. S8 and more discussions on page 9:

Figure S8. (a) Resistance of Au electrodes of tensile tests with the strain of 20% and 50% for 1000 cycles. (b) Resistance and length of Au electrodes of bend tests at angles up to 90 degree for 1000 cycles. (c) Resistance of Au electrodes as a function of strain. (d) Response current to NO with Au electrodes of different resistance. In a and b, $n = 3$ independent experiments.

“It is noted that as the sensor stretches, the resistance of Au electrodes changes over different strains (Fig. S8(c)). Different resistance of Au electrodes could affect the oxidation potential and response current for NO detection (Fig. S8(d)). It is therefore important to ensure that the resistance of Au electrodes does not change significantly during the course of NO detection.”

Reviewer #2

Summary Recommendation: This is an interesting paper dealing with a real time monitoring of nitric monoxide in living organism by an electrochemical sensor. In addition the parts of the sensor are claimed resorbable after certain time of being implanted in the mammal body. The originality level of the paper is high, however some important questions need to be addressed before it has been accepted for publication.

Our response: We thank the reviewer for these favorable comments.

- **Comment #1(a):** “What is the rationale behind using a biodegradable electrochemical sensor if it is integrated with data transmitting electronic device which is not biodegradable.”

Our response: We thank the reviewer for the comment. Biodegradable electrochemical sensor can serve as temporary diagnostic platforms for health assessment, pharmaceutical guidance, surgical intervention protocols, postsurgical monitoring, etc. Sensors can be implanted along with necessary surgery and resorbed or metabolized by the body after usage, eliminating a second surgery for device retraction and minimizing associated infection risks and hospital cost. A physically transient or biodegradable system would include both sensing and data transmitting parts. Both parts are novel electronic devices that need efforts to develop. In the current work, we realize a novel flexible and physically transient electrochemical sensor capable of continuous NO detection. A non-degradable transmitting electronic device at this stage can demonstrate advanced remote diagnostic functions of NO sensors. Future work would take a step forward to realize a fully transient data transmitting device, by leveraging the well-established CMOS foundry techniques, the potential feasibility of which has been shown by previous works. We added more discussions regarding the rationale behind using biodegradable electrochemical sensors.

Our modification to the manuscript: We added the following discussions on page 16-17:

“It is noted that a non-degradable wireless module at this stage can demonstrate advanced remote diagnostic functions of NO sensors. To achieve an entirely transient sensing platform, future efforts are needed to develop degradable data transmitting systems by leveraging well-established CMOS foundry techniques, the potential feasibility of which has been demonstrated by previous works [79, 80].”

- **Comment #1(b):** “Also the sensor contains metals (Au, Mo) which are non-biodegradable.”

Our response: We thank the reviewer for the comment. Biocompatibility and degradation studies of Au nanomaterials are mostly focused on nanoparticles, which have been proposed for various biomedical applications such as chemoradiation, photothermal therapy, drug delivery, et al. Despite of some contradictions, many reports suggest that gold nanoparticles are non-toxic with proper size and dosage, and metabolization occur through kidney, bladder or hepatobiliary system, e.g., no significant side effects were observed after 24 h with intravenous injection of gold nanoparticles ($\sim 100 \mu\text{g}/\text{rat}$). Although Au has been considered to be chemically inert, recent study reveals that gold nanoparticles (4-22 nm) are degraded *in vitro* by cells which is mediated by nicotinamide adenine dinucleotide phosphate (NADPH) oxidase in the lysosome, followed by a recrystallization process, indicating a potential metabolization mechanism for a trace amount of gold [54]. Our studies show that despite of the residual copper connection wires to the wireless module and the nondegradable sutures for implantation, the implanted NO sensor (substrates and Au electrodes) completely disappears after 8 weeks of implantation in the articular cavity (Fig. S17). Further evaluation of the element content of Mo and Au in the surrounding tissues of the sensor, liver, kidney and urine through inductively coupled plasma mass spectrometry (ICP-MS) suggest no detectable accumulation compared to that of the control group (Fig. S19). These results suggest that the NO device built with ultra-thin Au ($\sim 12.4 \mu\text{g}$) and poly(eugenol) ($\sim 0.3 \mu\text{g}$) layers on biodegradable PLLA-PTMC substrates is capable of full physical transience *in vivo* after 8 weeks of implantation,

through hydrolysis of PLLA-PTMC, disintegration of Au and poly(eugenol) nanomembranes and eventual potential clearance through phagocytosis and renal metabolization.

On the other hand, previous studies have revealed that Mo is biodegradable in aqueous environments and the reaction involved is $2\text{Mo} + 2\text{H}_2\text{O} + 3\text{O}_2 \rightarrow 2\text{H}_2\text{MoO}_4$. The recommended dietary allowance (RDA) for adult men and women is 45 $\mu\text{g}/\text{day}$. Mo has been used as dissolvable electrodes and interconnects for various transient electronics such as biodegradable batteries, intracranial pressure sensor, and neural sensors.

Our modification to the manuscript: We added the following discussions on page 6-8:

“Biocompatibility and degradation studies of Au nanomaterials are mostly focused on nanoparticles, which have been proposed for various biomedical applications such as chemoradiation, photothermal therapy, drug delivery, et al. [45-47]. Despite of some contradictions, many reports suggest that gold nanoparticles are non-toxic with proper size and dosage, and metabolization occur through kidney, bladder or hepatobiliary system [48-52], e.g., no significant side effects were observed after 24 h with intravenous injection of gold nanoparticles ($\sim 100 \mu\text{g}/\text{rat}$) [53]. Although Au has been considered to be chemically inert, recent study reveals that gold nanoparticles (4-22 nm) are degraded in vitro by cells which is mediated by nicotinamide adenine dinucleotide phosphate (NADPH) oxidase in the lysosome, followed by a recrystallization process, indicating a potential metabolization mechanism for a trace amount of gold [54]. Poly(eugenol) layer (thickness $\sim 16 \text{ nm}$) is incorporated to promote sensing selectivity and specificity toward NO by hydrophobic repulsion, ionic interaction and molecular exclusion [55]. Eugenol, the main chemical component of clove oil, has been used in dentistry for decades as an analgesic and demonstrated excellent biocompatibility [56]. Eugenol with an acceptable dietary intake (ADI) upper value of 2.5 mg/kg per day can be efficiently excreted by liver [57], and the lethal dosage LD_{50} of eugenol is reported to be 11 mg/kg in male F-344 rats and 17 mg/kg in

male Syrian golden hamsters [58]. Although there are few investigations on toxicity of poly(eugenol), it has been used in biosensors with desirable biocompatibility [41, 59]. The disintegration of ultrathin poly(eugenol) and possible degradation into eugenol could result in biocompatible products that can be metabolized by cells and organs [41, 60].”

“Previous studies have revealed that Mo is biodegradable in aqueous environments [62] and the recommended dietary allowance (RDA) for adult men and women is 45 µg/day [63]. Mo has been used as dissolvable electrodes and interconnects for various transient electronics such as biodegradable batteries, intracranial pressure sensors, and neural sensors [30, 33, 64].”

- ***Comment #1(c): “By the way, at the very end of the manuscript (Experimental Methods) the Authors state that the residual levels of these metals in body tissues/liquids were evaluated by ICP-MS technique but the corresponding data is not reported. Please provide it.”***

Our response: We thank the reviewer for pointing out the issue. The ICP-MS results were given in Figure S19 but we did not highlight that they are the ICP-MS results. The caption of Figure S19 is updated to clarify the confusion.

Our modification to the manuscript: We updated the caption of Figure S19:

“Figure S19: ICP-MS results showing Mo and Au concentrations of the tissues at the implantation site and various organs of the New Zealand Rabbit after 8 weeks. Sensor group (green): with NO sensor implantation. Control group (red): without implantation. (a) Mo concentration; (b) Au concentration. n = 3 independent experiments.”

- ***Comment #2: “There are some inconsistencies in the data presented on X-Y graphs. In particular, why the currents due to the addition of NO standards decay with time (Figs 2(b), (c)) while during interferences test they remain stable (Fig 2(f)). Additionally, L-arginine and L-NAME should be added to the list of tested interferents.”***

Our response: We thank the reviewer for pointing out the issue. The difference between Figs. 2(b)-(c) and Fig. 2f results from whether stirring is applied during data recording. To obtain the time dependent current response at different NO concentrations (calibration curves), stirring is required immediately after each addition of NO standard solutions to achieve uniformity. Data recording is then performed in the absence of stirring to ensure data stability especially at low NO concentrations ($<1 \mu\text{M}$, response current $< 11 \text{ nA}$), as stirring introduces significant noises which makes measurement impossible with ultra-low current signal. As a result, the recorded response current decays gradually at each test point, due to the sluggish diffusion of NO to supplement the NO consumption at the surface of the electrode. While for selectivity measurement, the added NO concentration is high ($100 \mu\text{M}$, response current $> 400 \text{ nA}$) and stirring can be continuously applied with limited influence on the data stability, which contributes to the sustained response current (Fig. 2f). More descriptions are added in the manuscript to clear the confusion.

The influence of L-Arginine and L-NAME on NO detection are given in Fig. S12(a). When L-Arginine and L-NAME are added into PBS solutions, there is no current response, indicating that they are not interfering with the NO detection.

Our modification to the manuscript: We added the following descriptions on page 10-11:

“During the course of the measurement, stirring is required upon each addition of NO standard solutions to achieve uniformity, followed by data recording in the absence of stirring to minimize noises and ensure data stability especially at low NO concentrations ($<1 \mu\text{M}$, response current $< 11 \text{ nA}$). As the NO concentration increases, an increase in the current response can be rapidly captured ($< 350 \text{ ms}$), which is important for real-time NO monitoring. The subsequent current attenuation is mainly attributed to the relatively sluggish diffusion of NO to the electrode surface in the PBS.”

“Given the additions of high concentrations of NO and interfering chemicals, stirring can be continuously applied to achieve uniformity and yet maintain stable response current.”

The corresponding methodology was also updated on page 22:

“To acquire an accurate and stable response current signal for the NO calibration curve, especially at low concentrations, mechanical stirring was applied to achieve a uniform NO concentration before recording the response current, and then stirring was turned off during the short period of data recording.”

“For selectivity tests, NO and interfering chemicals (glucose, sodium nitrite, sodium nitrate, ascorbic acid, and uric acid) were added in sequence in PBS, and the response current was recorded with mechanical stirring.”

Fig. S12(a) is added in the supporting materials.

Figure S12. (a) Selectivity measurement in PBS: current response with the additions of L-Arg (5 mM) and L-NAME (10 mM) and NO solutions (0.1 mM).

We added the corresponding description on page 13:

“The results of interfering tests of L-Arg and L-NAME are given in Fig. S12(a).”

The corresponding methodology was also updated on page 22:

“Interfering tests were also performed on L-Arg (5 mM) and L-NAME (10 mM).”

- **Comment #3:** *“The list of references should be updated as follows: Ref [41] has nothing to*

do with poly(eugenol). Please update accordingly. Ref [46] is not adequate to the sequence it is listed in. It should be rather replaced by the very first paper reporting the application of poly(eugenol) as an NO-selective membrane i.e. Ciszewski A., Milczarek G. *Electroanalysis*, [10] 791 (1998).”

Our response: We thank the reviewer for pointing out the issue. Ref [41] has been updated and Ref [46] has been replaced by the very first paper reporting the application of poly(eugenol) as a NO-selective membrane.

Our modification to the manuscript: The Ref. [41] has been replaced by the following reference:

Quinton, D., et al. On-chip multi-electrochemical sensor array platform for simultaneous screening of nitric oxide and peroxynitrite. *Lab Chip* 11, 1342-1350 (2011).

Ref. [46] (Ref. [55] in revised manuscript) has been replaced by the following reference: Ciszewski, A. & Milczarek, G. A New Nafion-Free Bipolymeric Sensor for Selective and Sensitive Detection of Nitric Oxide. *Electroanal* 10, 791-793 (1998).

Reviewers' Comments:

Reviewer #1:

Remarks to the Author:

All comments from the reviewer were well addressed in the revised manuscript.